# Uncertainty Quantification in Segmenting Tuberculosis-Consistent Findings in Frontal Chest X-rays

**DOI:** 10.3390/biomedicines10061323

**Published:** 2022-06-04

**Authors:** Sivaramakrishnan Rajaraman, Ghada Zamzmi, Feng Yang, Zhiyun Xue, Stefan Jaeger, Sameer K. Antani

**Affiliations:** National Library of Medicine, National Institutes of Health, Bethesda, MD 20892, USA; ghadazamzmi.alzamzmi@nih.gov (G.Z.); feng.yang2@nih.gov (F.Y.); zhiyun.xue@nih.gov (Z.X.); stefan.jaeger@nih.gov (S.J.); santani@mail.nih.gov (S.K.A.)

**Keywords:** chest X-ray, uncertainty, uncertainty quantification, deep learning, medical image segmentation, tuberculosis, confidence intervals, Monte–Carlo Dropout, mean average precision

## Abstract

Deep learning (DL) methods have demonstrated superior performance in medical image segmentation tasks. However, selecting a loss function that conforms to the data characteristics is critical for optimal performance. Further, the direct use of traditional DL models does not provide a measure of uncertainty in predictions. Even high-quality automated predictions for medical diagnostic applications demand uncertainty quantification to gain user trust. In this study, we aim to investigate the benefits of (i) selecting an appropriate loss function and (ii) quantifying uncertainty in predictions using a VGG16-based-U-Net model with the Monto–Carlo (MCD) Dropout method for segmenting Tuberculosis (TB)-consistent findings in frontal chest X-rays (CXRs). We determine an optimal uncertainty threshold based on several uncertainty-related metrics. This threshold is used to select and refer highly uncertain cases to an expert. Experimental results demonstrate that (i) the model trained with a modified Focal Tversky loss function delivered superior segmentation performance (mean average precision (mAP): 0.5710, 95% confidence interval (CI): (0.4021,0.7399)), (ii) the model with 30 MC forward passes during inference further improved and stabilized performance (mAP: 0.5721, 95% CI: (0.4032,0.7410), and (iii) an uncertainty threshold of 0.7 is observed to be optimal to refer highly uncertain cases.

## 1. Introduction

The 2021 World Health Organization (WHO) Global Tuberculosis (TB) report [1] mentions that roughly a quarter of the global population is infected with TB, a disease caused by the *Mycobacterium tuberculosis* bacteria. Approximately 1.5 million TB-related deaths were reported in 2020. There was a 7% increase in TB-related deaths for the first time in more than a decade due to the disruption in access to medical care as well as socio-economic factors. Many services were reallocated from tackling TB to responding to the COVID-19 pandemic. This increase makes it imperative to develop automated and reliable early TB screening and diagnostic methods. Chest X-ray (CXR) imaging remains the most widely used imaging modality for TB screening [2,3,4,5,6]. However, there is a lack of human expertise in interpreting the images, particularly in low resource regions [7,8] that are also the ones most significantly impacted by the disease. Automated TB-consistent region segmentation methods could play an important role in developed solutions [9,10].

Deep learning (DL)-based methods have demonstrated superior performance in medical image segmentation tasks. The loss function used during model training helps measure the goodness of the model predictions. The choice of the loss function is directly related to the data characteristics [11]. Manually segmenting the disease regions of interest (ROIs) currently serves as the gold standard for evaluating the segmentation performance. There is vast heterogeneity in the visual characteristics of TB-consistent findings. Further, the ROIs are a relatively small portion of the whole image. This may lead to biased learning of the majority background pixels over the minority ROI pixels, thereby adversely impacting segmentation and generalization performance. While generalized loss functions, e.g., cross-entropy and means squared error [12], could be used, it is preferable to use specialized loss functions [13] that would resolve the bias in learning the minority disease positive (ROI) pixels. To the best of our knowledge, except for [10] no published literature evaluates the selection of appropriate loss functions, particularly for segmenting TB-consistent regions in CXRs. Appropriately selected loss functions can significantly impact prediction quality, making their evaluation a critical step in model training.

A principal limitation of DL methods is that they require huge amounts of labeled data to deliver reliable outcomes. Obtaining a large number of medical images and their associated disease-specific masks are challenging due to several factors, including varying image acquisition protocols resulting in differing quality, the variability in the disease pathogenicity in different parts of the world which is reflected through radiological signs of TB-consistent manifestations, insufficiently or weakly labeled data sets [14], and their availability for research. Studies in the literature [3,14] used CXRs showing TB-consistent manifestations accompanied by coarse annotations in the form of rectangular bounding boxes. Such annotations implicitly introduce errors in model training, since a fraction of non-TB-consistent region pixels are considered as the positive class. This fraction of *false-positive* pixels is dependent on the inter-and intra-reader annotation variability. To the best of our knowledge, we are not aware of any publicly available CXR datasets that include fine-grained (pixel-wise) annotations of TB-consistent regions. 

Another limitation of DL models is that they do not explain uncertainty in their predictions [15]. Uncertainty quantification is indispensable, particularly considering healthcare applications to (i) explicitly identify and handle uncertain outputs and (ii) check for reliability in model outcomes. The Softmax outputs are not a reliable measure of confidence since a poorly calibrated model could often result in overconfident predictions [16,17]. Additionally, one must not disregard challenges due to inter-and intra-reader variability in the expert annotations that may lead to uncertain predictions [18]. Recently, several soft label-based methods [19] have been incorporated into the loss functions to address labeling uncertainty. For example, the authors of [16] used morphological operators such as dilation and erosion to restrict the soft labels to the ROI boundaries. Such an approach ensured appropriately representing labeling uncertainty and improved model robustness to segmentation errors. Methods such as Bayesian learning and ensemble learning have been proposed to provide uncertainty measures [20,21]. Bayesian learning can be incorporated into segmentation to provide a measure of uncertainty and weight regularization [22]. A simple but effective method of implementing Bayesian learning is called the Monte–Carlo Dropout (MCD) method, implemented in [23], which formulates (i) the conventional dropout as an equivalent to Bayesian variational inference and (ii) integrates over models’ weights to provide uncertainty estimates without increasing the computational complexity or sacrificing performance. Several uncertainty metrics, such as aleatoric uncertainty [24], epistemic uncertainty [24], and entropy [25], can be used to quantify uncertainties. An uncertainty threshold could therefore be derived from these metrics that would help identify the most uncertain cases and refer them to the human expert for consideration. Such analysis is critical for (i) identifying individual uncertain cases rather than providing an aggregated performance measure using the entire test set, and (ii) providing reliable, safe, outcomes and allowing the optimized use of resources. 

The main contributions of this study are summarized as follows:We conduct extensive empirical evaluations to select an appropriate loss function to improve model performance.The proposed method quantifies uncertainty in predictions using the MCD method and identifies the optimal number of MC samples required to stabilize model performance.We evaluate, quantify, and compare various uncertainties in model representations and arrive at an optimal uncertainty threshold using these uncertainty metrics. The predictions exceeding this threshold could be referred to an expert to ensure reliable outcomes.To the best of our knowledge, this is the first study that uses fine-grained annotations of TB-consistent regions for model training and evaluation.

Section 2 elaborates on the materials and methods, Section 3 discusses the results, and Section 4 concludes the study.

## 2. Materials and Methods

### 2.1. Data Collection and Preprocessing

The Shenzhen TB dataset published in [3] is used to train and evaluate the VGG16-based U-Net model. The collection includes 336 de-identified CXRs captured from TB patients and 326 CXRs with no abnormal findings. The cases in the dataset have been confirmed by culture, and that typical TB appearance in imaging combined with a positive response to anti-TB medication was a criterion for confirming TB. The patients may have a history of TB with some images showing secondary TB. All images were taken close to an active infection. The CXRs vary in dimensions but are approximately 3000 × 3000 pixels. The abnormal CXRs are annotated by an experienced radiologist using the Firefly (https://cell.missouri.edu/software/firefly/, accessed on 3 December 2021) annotation tool for 19 abnormalities, viz., *pleural effusion*, *apical thickening*, *single nodule* (non-calcified), *pleural thickening* (non-apical), *calcified nodule*, *small infiltrate*, *cavity*, *linear density*, *severe infiltrate* (consolidation), *thickening of the interlobar fissure*, *clustered nodule*, *moderate infiltrate*, *adenopathy*, *calcification* (other than nodule and lymph node), *calcified lymph node*, *miliary*, *retraction*, *other*, and *unknown*. We used 17 abnormalities to train the models; the CXRs with only the *unknown* and *other* labels are included in the test set. These annotations are stored in TXT format and then prepared in JSON format for processing as well as preparing separate binary mask images for each abnormal area. The radiological signs consistent with TB are observed only in 330 CXRs. 

Preprocessing: We (i) combined the masks to include all TB-consistent abnormalities for a given CXR, (ii) binarized the masks such that the pixels corresponding to the disease-positive (ROI) TB-consistent manifestations are set to 1, and pixels corresponding to the negative (background) are set to 0, and (iii) resized the masks and CXRs to 256 × 256 spatial dimensions. The dataset is divided at the patient level into the train, validation, and test sets, as shown in Table 1, to prevent data leakage and biased training. The training images are further augmented using the Augmentor tool [26] to perform transformations such as rotation [−12, 12], horizontal flipping, zooming [0.8, 1.2], contrast change [1, 1.1], brightness change [1, 1.1], histogram equalization, and elastic distortion with a magnitude of 8 and a grid size of (4, 4) to produce 2000 additional CXRs and their associated masks; a sample is shown in Figure 1.

### 2.2. Statistical Analysis

We evaluated statistical significance using the mean average precision (mAP) metric achieved by the models trained with various loss functions. Statistical evaluation was performed using the 95% confidence interval (CI), which is measured as the binomial interval using the Clopper–Pearson method. We followed [27] to obtain the p-value from the CI. The standard error (S) is measured from the lower (*l*) and upper (*u*) limits of the 95% CI, as shown in Equation (1).
(1)S=(u−l)(2×1.96)

The value of 1.96 denotes that 95% of the area under the normal distribution curve would lie within 1.96 standard deviations away from the mean value (*µ*). The test statistic (z) is computed as shown in Equation (2).
(2)$$z=\frac{Difference}{S}$$

Here, *Difference* denotes the estimated differences in the measured mAP metric between the segmentation models trained with various loss functions. We calculate the *p*-value from the test statistic z, as shown in Equation (3).
(3)$$p=exp\left(-0.717\times z-0.416\times z^{2}\right)$$

### 2.3. TB-Consistent Region Segmentation

#### 2.3.1. Model Architecture

A U-Net model with a VGG-16 [28] encoder (Figure 2) is used to segment TB-consistent regions. The VGG-16-based encoder is initialized with ImageNet weights.

The encoder is made up of five convolutional blocks. The first block consists of two convolutional layers with ReLU activation and 64 filters. The number of filters increases by a factor of 2 in the succeeding convolutional blocks. The downward red arrows represent the max-pooling layers after each convolutional block. The decoder consists of five convolutional blocks. Each convolutional layer in the first, second, third, and fourth convolutional block is followed by batch normalization, ReLU activation, and dropout (rate = 0.3) layers. The upward green arrows denote up-sampling operations performed after each convolutional block to restore the images to their original input dimensions. The grey arrows denote concatenate operations performed to combine the spatial information from the encoder (down-sampling) path with the decoder (up-sampling) path to retain good spatial information. The final convolutional layer with sigmoidal activation predicts the binary masks for a given input. The proposed U-Net model is trained and validated on the Shenzhen TB CXRs (from Table 1) using an Adam optimizer with an initial learning rate of 1 × 10^−4^ and a batch size of 16. We used callbacks to store model checkpoints. Early stopping is used to stop training the model when no improvement in the validation loss is observed in the last 20 epochs. The model weights that deliver superior performance with the validation data are used to predict the hold-out test data and the performance is recorded.

#### 2.3.2. Loss Functions

We evaluated the segmentation performance while training the VGG-16-based U-Net model using the following loss functions: (i) Cross-entropy (CE) loss; (ii) CE + Boundary uncertainty (BU) loss (iii) Dice loss; (iv) Dice + BU loss; (v) Intersection of Union (IOU) loss; (vi) IOU + BU loss; (vii) Tversky loss; (viii) Tversky + BU loss; (ix) Focal Tversky loss; and (x) Focal Tversky + BU loss. 

The CE loss is the most commonly used loss function in image segmentation. It is computed as shown in Equation (4). Here, we have two probability distributions: The predictions can either be ℙ (Y’ = 0) = $y^{\prime }$ or ℙ (Y’ = 1) = 1 − $y^{\prime }$. The GT can either be ℙ (Y = 0) = y or ℙ (Y = 1) = 1 − y where *y* ∈ {0, 1}. The predictions are given by the Sigmoid function shown in Equation (5) for an input *x*.
(4)$$BCE_{loss}\left(y,y^{\prime }\right)=-(ylog\left(y^{\prime }\right)+\left(1-y\right)\log\left(1-y^{\prime }\right)$$
(5)$$y\prime =1/\left(1+e^{-x}\right)$$

The *IOU*/Jaccard score and the Dice index/F1 score are other widely used metrics to evaluate segmentation performance [10]. Let *TP*, *FP*, and *FN* denote the true positives, false positives, and false negatives, respectively. Given a pre-defined *IOU* threshold, a predicted mask is considered to be *TP* if it overlaps with the *GT* mask by a value exceeding this threshold. *FP* denotes that the predicted mask has no associated *GT* mask. *FN* denotes that the *GT* mask has no associated predicted mask. Then, the *IOU* and *IOU* loss values are computed as shown in Equations (6) and (7). 

The Dice index and Dice loss values are given by Equations (8) and (9). Similar to the *IOU* metric, the value of the Dice index ranges from [0, 1]. Higher values of the Dice index denote improved similarity between the predicted and *GT* masks.
(6)IOU (Jaccard Score)=TP/(TP+FP+FN)
(7)$$IOU_{loss}=1-IOU$$
(8)Dice (F1−score)=2×TP/(2×TP+FP+FN)
(9)$$Dice_{loss}=1-Dice$$

Another loss function called the Tversky loss is constructed from the Tversky index (*TI*) function [13], a generalization of the Dice index. The *TI* function adds weight to *FP* and FN and is expressed as shown in Equations (10) and (11). Here, c denotes the minority disease-positive (ROI) class. When β = 0.5, the equation simplifies to the regular Dice index.
(10)$$TI\left(y,y^{\prime }\right)=\frac{yy^{\prime }}{yy^{\prime }+\beta \left(1-y\right)y^{\prime }+\left(1-\beta \right)y\left(1-y^{\prime }\right)}$$
(11)$$TI_{loss}\left(y,y^{\prime }\right)=\underset{c}{\sum }1-TI{\left(y,y^{\prime }\right)}_{c}$$

The Focal Tversky (*FT*) loss function [13] is a generalized focal loss function based on the *TI*. It is parameterized by *γ* to balance between the majority negative (background) and minority disease-positive (ROI) pixels. The *FT* loss is given by Equation (12). After empirical evaluations, we fixed the value of *β* = 0.7 and *γ* = 0.75.
(12)$$FT_{loss}{\left(y,y^{\prime }\right)}_{c}=\underset{c}{\sum }1-TI_{c}^{\gamma }$$

In a binary segmentation problem, each pixel *k* in the *GT* mask, at location *t*, is assigned a hard class label as shown in Equation (13).
(13)$$k_{t}:\{\begin{array}{c} k=1,if\;t\in \tau \\ k=0,if\;t\notin \tau \end{array}$$

Here, τ denotes the target. In the boundary uncertainty (BU) approach, proposed by [29], the hard labels 0 and 1 are converted into soft labels to represent probabilistic scores as shown in Equations (14) and (15).
(14)$$k_{t}:\{\begin{array}{c} k\leq 1,if\;t\in \tau \\ k\geq 0,if\;t\notin \tau \end{array}$$
(15)$$k_{t\notin \tau }\leq k_{t\in \tau }$$

Equation (14) denotes that the values closer to 1 and 0 denote higher confidence in classifying the pixels as belonging to the ROI or background, respectively. The soft labels are restricted only to the ROI boundaries to approximate the uncertainty in manual segmentation using morphological operators such as dilation (⨹) and erosion (⨺). Let *X* denote the input image of dimension *a × b*. The BU function performs dilation (⨹) and erosion (⨺) operations on the ROI boundaries at all positions by querying with a structural element *Y* of 3 *×* 3 spatial dimensions, as shown in Equations (16) and (17). Probabilities are then assigned for the pixels on the ROI boundaries, as shown in Equation (18).
(16)$$\left(X⨹Y\right)\left(x,y\right)=max_{i\in S1j\in S2}\left(X\left(x-i,y-j\right)+Y\left(i,j\right)\right)$$
(17)$$\left(X⨺Y\right)\left(x,y\right)=min_{i\in S1j\in S2}(X\left(x+i,y+j\right)-Y\left(i,j\right))$$
(18)$$k_{t\in \tau }:\{\begin{array}{c} k=\alpha ,if\;k\in (\left(X⨹Y{)}_{n}-X\right)\\ k=\beta ,if\;k\in (X-\left(X⨺Y{)}_{n}\right) \end{array}$$

Here, *n* = 1 denotes the number of iterations for which the erosion and dilation operations are performed. The hyperparameters *α* and *β* denote the values for the soft labels that are exterior and interior to the ROI boundaries, respectively. When *α* = 1 and *β* = 0, the soft labels would converge to the original hard labels. After empirical evaluations, we fixed the value of *α* = 0.6 and *β* = 0.4. We incorporated the BU with BCE, Dice, *IOU*, Tversky, and Focal Tversky losses to evaluate for an improvement in performance compared to using hard labels. 

The *mean average precision* (*mAP*) is measured as the area under the precision-recall curve (AUPRC), as shown in Equation (19). The precision measures the accuracy of predictions and recall measures of how well the model identifies all the *TP*s. They are computed as shown in Equations (20) and (21). The value of mAP lies in the range [0, 1].
(19)$$mean\;average\;precision\;\left(mAP\right)=\int_{0}^{1}Precision\left(Recall\right)dRecall$$
(20)$$Precision=\frac{TP}{TP+FP}$$
(21)$$Recall=\frac{TP}{\left(TP+FN\right)}$$

#### 2.3.3. Uncertainty Quantification

Uncertainty in predictions is measured to demonstrate the confidence of the model about the class of each pixel, viz. TB-consistent abnormality (*positive*) or background (*negative*), in the predicted mask. Monte–Carlo Dropout (MCD) [23], a method of Bayesian learning and inference, is commonly used to estimate this uncertainty. Dropout is conventionally used during model training to prevent overfitting of the training data and improve generalization [30]. However, dropout can also be used during inference to produce an ensemble prediction without additional computational cost. The idea behind the MCD approach is to keep the dropout layers active during both model training and inference. It is a simple and efficient method for implementing Bayesian learning, where a variational posterior distribution is defined for the weight matrices, as shown in Equations (22) and (23).
(22)$$X_{i}\sim Bernoulli\;\left(b_{i}\right)$$
(23)$$W_{i}=M_{i}.diag\left(X_{i}\right)$$

Here, $X_{i}$ denotes the coefficients of random activation or inactivation, $W_{i}$ and $M_{i}$ denotes the weight matrices after and before applying dropouts, respectively, and $b_{i}$ denotes the probability for activation for a given layer *i*. During inference, we perform *N* forward passes through the model. That is, we predict the outcome for a given input *N* times to approximate the Bayesian variational inference and the results are aggregated. During each forward pass, a different set of model units are sampled to be dropped out, which results in stochastic predictions that can be interpreted as samples from a probabilistic distribution [23]. We quantify the uncertainty in the predictions using the probabilistic samples during *N* forward passes by evaluating the following uncertainty metrics. 

Aleatoric uncertainty [24]: This uncertainty is an inherent property of data distribution that may arise due to a class overlap or the presence of noise in the data. It is expressed as shown in Equations (24) and (25).
(24)Aleatoric=1/N∑n=1Ndiag(jn^)−(jn^jn^)N
(25)jn^=Softmax(fwnx)

Here, $f_{w_{n}}$ denotes the last pre-activated linear output of the model and $w_{n}$ for *n* = {*1* to *N*) are the weights sampled randomly from the variational distribution with a pre-defined sampling number *N* for the prediction y given an input x. 

Epistemic uncertainty [24]: This uncertainty arises due to limited data and insufficient knowledge about the model. This may be because of the difference between the train and test distribution. It is expressed as shown in Equations (26) and (27). Epistemic uncertainty can be reduced by collecting more training data or optimizing the models.
(26)Epistemic=1/N∑n=1N(jn^−jn′)(jn^−jn′)N
(27)jn′=1/N∑n=1Njn^

Total uncertainty: This combined measure of aleatoric and epistemic uncertainty scores could be used to estimate if both the model and data point themselves are uncertain. It is the union of all uncertainty-contributing pixels and is expressed as shown in Equation (28).
(28)Total uncertainty=Aleatoric+Epistemic

Entropy [31]: Entropy is the measurement of randomness or disorder in processed information and is expressed as shown in Equation (29).
(29)$$H=-\underset{y\in Y}{\sum }Z\left(\frac{y}{x}\right)log_{2}Z\left(\frac{y}{x}\right)$$

Here, $Z\left(\frac{y}{x}\right)$ is the output *y* from the Softmax layer given an input *x.* The value of entropy ranges from 0 to 1, where 0 denotes that the data is perfectly classified and 1 denotes complete randomness. The units of entropy depend on the base (b) of the logarithm used. Here, we used b = 2; in this case, the entropy is expressed in Shannon/bits.

We identified the optimal number of forward (*N*) passes during inference that stabilizes performance. We further identified an optimal uncertainty threshold (*τ*) based on the aforementioned uncertainty metrics. The predictions exceeding the value of *τ* are referred to an expert. This ensures that the model predicts masks only for cases where it is certain, while alerting the clinician to uncertain predictions. The technique could be used to develop clinician confidence in the prediction model, while simultaneously reducing verification effort.

## 3. Results

We organized the results into the following sections: (i) Evaluating the performance of the VGG-16-based U-Net model trained with the proposed loss functions (Section 3.1), (ii) quantifying uncertainty in predictions (Section 3.2), and (iii) identifying the optimal uncertainty threshold (Section 3.3). 

### 3.1. Segmentation Performance Achieved with the Proposed Loss Functions

Recall that the VGG-16-based U-Net model is trained using various loss functions as discussed in Section 2.3.2. Table 2 lists the performance achieved by the model with the hold-out test set and Figure 3 shows the performance curves. 

From Table 2, we observe that the VGG-16-based U-Net trained with the Focal Tversky + BU loss demonstrated superior values in terms of *mAP*, *IOU*, and Dice metrics. The model trained with the Focal Tversky loss demonstrated superior values regarding the AUC metric. The 95% CI for the mAP metric achieved by the model trained with Focal Tversky + BU loss demonstrated a tighter error margin and hence higher precision and is observed to be significantly superior (*p* < 0.05) to those achieved by the models trained with CE and Dice + BU losses. Since higher values of mAP denote that the model can better handle the positive (ROI) class and is more accurate in its detection, the model trained with the Focal Tversky + BU loss is chosen for further analysis. Figure 4 shows a sample CXR from the hold-out test set, the ground-truth (*GT*) mask, and the predictions of the models trained with the proposed loss functions. From Figure 4, we observe that predictions from the model trained with the Focal Tversky + BU loss closely matched the *GT*, followed by Tversky, and Dice losses.

### 3.2. Uncertainty Quantification 

We observe the uncertainty in predictions using the aforementioned metrics, viz., aleatoric, epistemic, entropy, and plotted them over the number of MC forward (*N*) passes. The mean and standard deviation of these uncertainty values are recorded. We also measured the elapsed time (in seconds). Figure 5 shows the variation in these uncertainty metrics with the number of MC forward passes, ranging from 1 to 50. 

It is observed from Figure 5 that the variance in the uncertainty decreases (illustrated with the error bars length) with an increase in the number of MC forward passes. The entropy, aleatoric, and epistemic uncertainties are observed to vary until 30 MC forward passes and then begins to stabilize. This indicates that *N* = 30 would be a good stabilizing point that can be used to evaluate the TB-consistent region segmentation performance. Table 3 compares the performance of the baseline (Focal Tversky + BU) model with those achieved by averaging the predictions achieved with 30 MC forward passes. We observed from Table 3 that the performance achieved with 30 MC forward passes is superior, though not significantly different (*p* > 0.05) in terms of the mAP, IOU, and Dice metrics, compared to the baseline. 

We visualized how the different uncertainties are conveyed in the predictions. Figure 6 shows the results of segmentation representing these uncertainties for an instance of CXR from the hold-out test set. 

It is observed from Figure 6 that the uncertainty is commonly observed along the predicted boundaries of the ROIs consistent with TB. Predicted pixels with lower uncertainties look “unfilled” while those with greater uncertainties appear prominent, as shown in Figure 6. We also observed that the total uncertainty and entropy appear similar, providing a fine-grained representation of boundary region pixels with greater uncertainty.

### 3.3. Identifying the Optimal Uncertainty Threshold 

We identified the optimal uncertainty threshold (*τ*) using each of the aleatoric, epistemic, and total uncertainties. The optimal threshold (*τ*) is defined as the maximum permissible uncertainty exceeding which the individual sample shall be referred to an expert. Such an estimation would help minimize the human effort and clinical workload, thereby allowing the expert to only look into the most uncertain cases. We fixed the experimental range of *τ* to [0.1, 0.9] with a step size of 0.1. The performance metrics including accuracy, precision, Dice, mAP, AUC, and IOU are calculated for varying values of *τ,* as shown in Figure 7, Figure 8 and Figure 9. From Figure 7, Figure 8 and Figure 9, we observe that with increasing *τ*, the accuracy, precision, Dice, mAP, AUC, and IOU remain steady or do not notably change until τ = 0.7. Thereafter, the performance decreases as the model begin to predict highly uncertain cases. A similar pattern is observed for all uncertainty metrics. Thus, a value of *τ* = 0.7 is identified to be optimal for segmenting TB-consistent regions using the dataset under study. This ensures maximizing performance while automatically referring only the highly uncertain cases to an expert allowing for reliable outcomes. 

## 4. Discussion and Conclusions

Recall that the CXR images with *unknown* and *other* labels are included in the hold-out test set for evaluating segmentation performance. These classes are not observed by the model during training. We visualized the predictions of the model trained with the Focal Tversky + BU loss for instances of test CXRs with *unknown* and *other* labels, as shown in Figure 10. We observed that the model performed fairly well with its predictions for CXRs with *unknown* and *other* labels. These findings suggest that the learned features generalized well to images with *unknown* and *other* class labels, which suggest that they are consistent with TB and exhibit similar radiological signs. 

The selection of optimal threshold *τ* for segmenting TB-consistent regions depends on the characteristic of the data under study. For segmenting TB-consistent regions, it is sensible to identify the optimal *τ* based on expert guidance and the application. The threshold shall be chosen in a way to refer minimal cases to the expert and reduce the clinical workload, while also segmenting only the most certain cases to provide reliable predictions.

This study, however, suffers from the limitation that the CXRs manifesting TB-consistent lesions that are used to train and evaluate the segmentation models are limited. Additional diversity in the training process could be introduced by using CXR data and annotations from cross-institutions. Next, novel model optimization methods and loss functions can be proposed to further minimize prediction uncertainty and improve confidence. It is to be noted that more recent architectures such as SegNet [32] and Trilateral attention net [33] could have been used in this study. However, the objective of this study is not to propose a new model or demonstrate state-of-the-art results for TB-consistent region segmentation. Rather, it is to validate the use of appropriate loss functions suiting the data under study and quantify uncertainty in model representations. Any DL model architecture could be modified to establish Bayesian variational inference using the MCD method for uncertainty quantification. These models that provide predictive estimates of uncertainty could be widely integrated into clinical practice to flag clinicians for alternate opinions, thereby imparting trust, and improving patient care. Research is ongoing in proposing novel methods for quantifying and explaining uncertainties in model predictions [34,35]. With the advent of high-performance computing and storage solutions, several models with deep and diverse architectures can be trained to construct ensembles with reduced prediction uncertainty and deployed in the cloud to be used for real-time clinical applications. This study, however, serves as a paradigm to quantify uncertainties and establish an uncertainty threshold for segmenting disease-specific ROIs in CXRs and could be extended to other medical imaging modalities. Future studies will focus on quantifying uncertainties in multiclass and multi-label medical data classification, regression, segmentation, and uncertainty-based active learning.

## Figures and Tables

**Figure 1 biomedicines-10-01323-f001:**
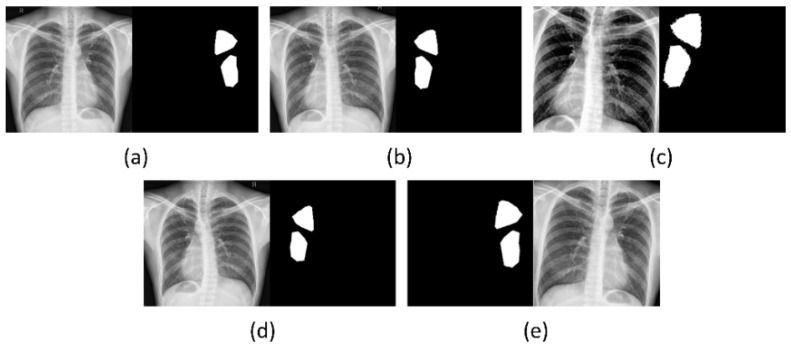
Augmented samples for a CXR instance from the training set. (**a**) Original CXR and its associated TB-consistent region mask; (**b**) horizontal flipping; (**c**) horizontal flipping, contrast change, and zooming; (**d**) elastic distortion, and (**e**) zooming and clockwise rotation.

**Figure 2 biomedicines-10-01323-f002:**
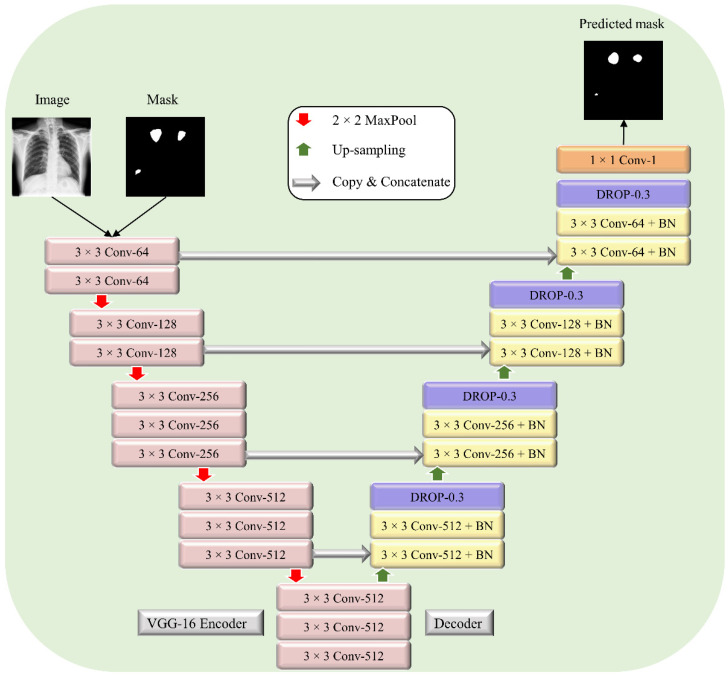
The architecture of the proposed VGG-16-based U-Net model. The encoder is a VGG-16 model initialized with ImageNet weights.

**Figure 3 biomedicines-10-01323-f003:**
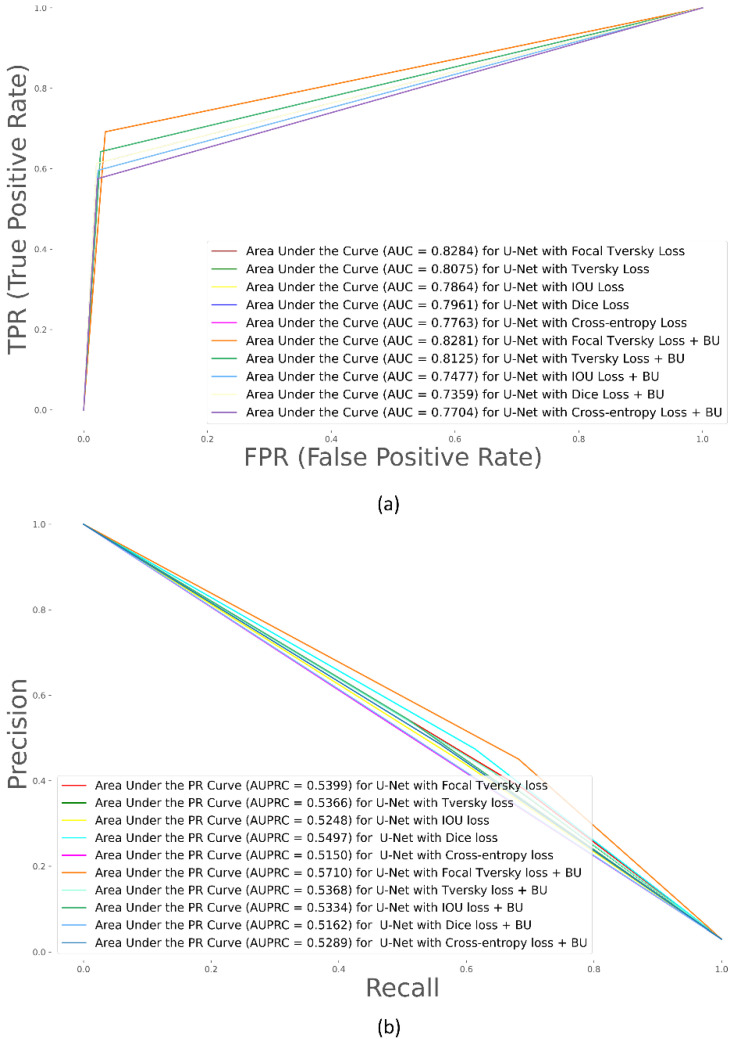
Test performance curves for the VGG-16-based U-Net model trained with various loss functions. (**a**) ROC and (**b**) PR curves. The area under the PR curve (AUPRC) gives the mAP values. The AUC and AUPRC curves shown in brown color denote superior performance achieved by the models trained with Focal Tversky and Focal Tversky + BU losses, respectively.

**Figure 4 biomedicines-10-01323-f004:**
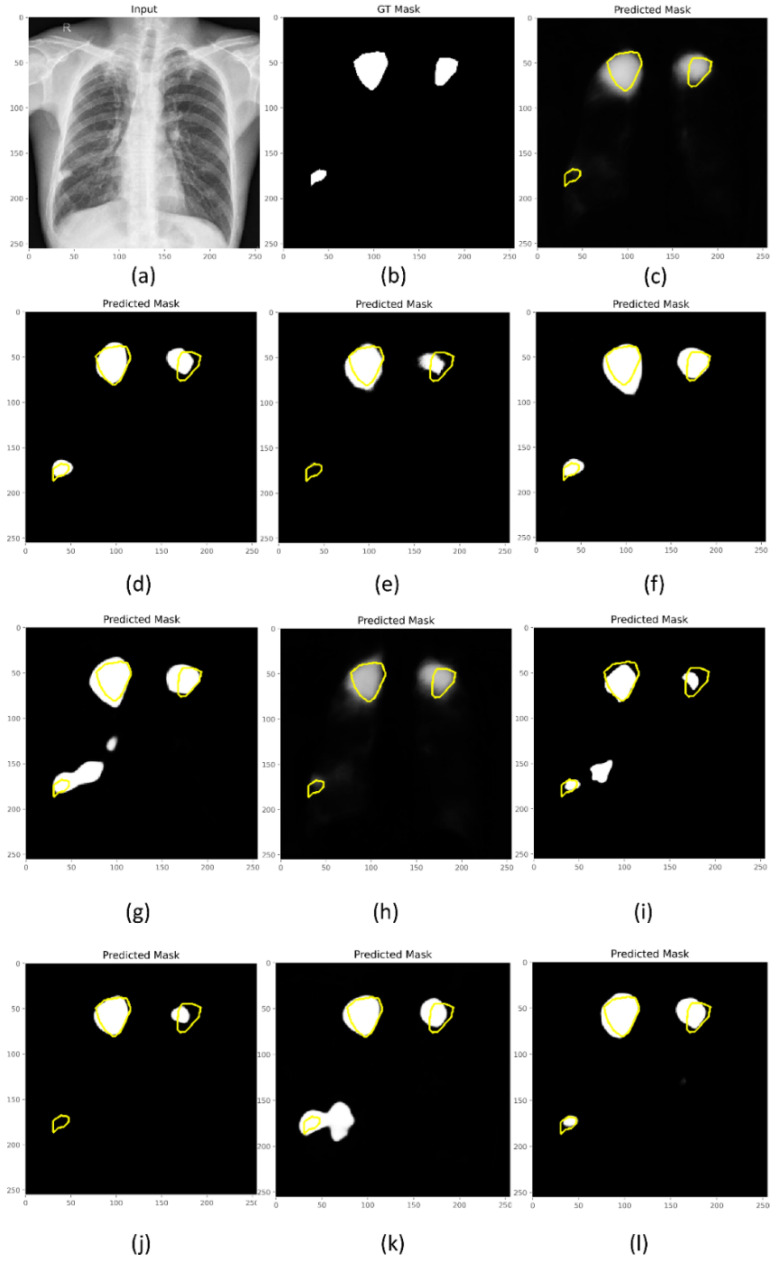
Predictions of the VGG-16-based U-Net model for a sample CXR. (**a**) Input CXR; (**b**) GT mask; Losses without BU (**c**) CE; (**d**) Dice; (**e**) IOU; (**f**) Tversky; (**g**) Focal Tversky; losses with BU (**h**) CE; (**i**) Dice; (**j**) IOU; (**k**) Tversky, and (**l**) Focal Tversky. The pixels in yellow denote the contour of the GT masks.

**Figure 5 biomedicines-10-01323-f005:**
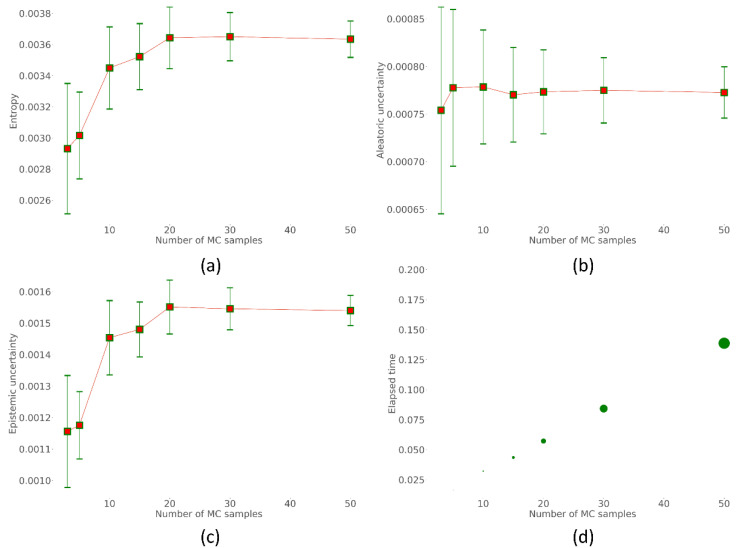
Identifying stabilizing points for various uncertainties based on the number of MC forward (*N*) passes. (**a**) Entropy; (**b**) aleatoric uncertainty; (**c**) epistemic uncertainty, and (**d**) elapsed time (in seconds).

**Figure 6 biomedicines-10-01323-f006:**
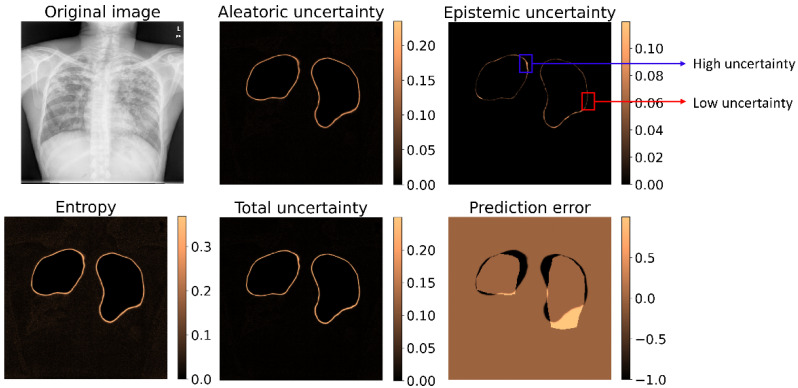
Uncertainty representation using various uncertainty metrics and the prediction error are shown for a sample CXR from the test set. The blue bounding box denotes predicted pixels with high uncertainty, and the red bounding box shows those with low uncertainty.

**Figure 7 biomedicines-10-01323-f007:**
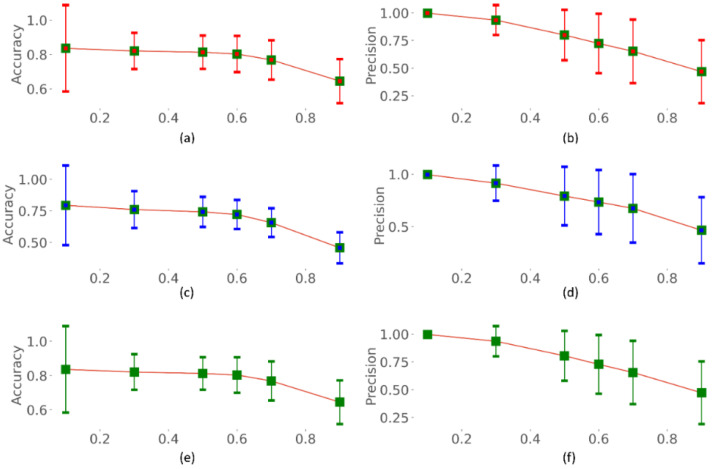
Accuracy and precision measured for varying values of *τ*. (**a**,**b**) using aleatoric uncertainty; (**c**,**d**) using epistemic uncertainty; and (**e**,**f**) using total (aleatoric + epistemic) uncertainty.

**Figure 8 biomedicines-10-01323-f008:**
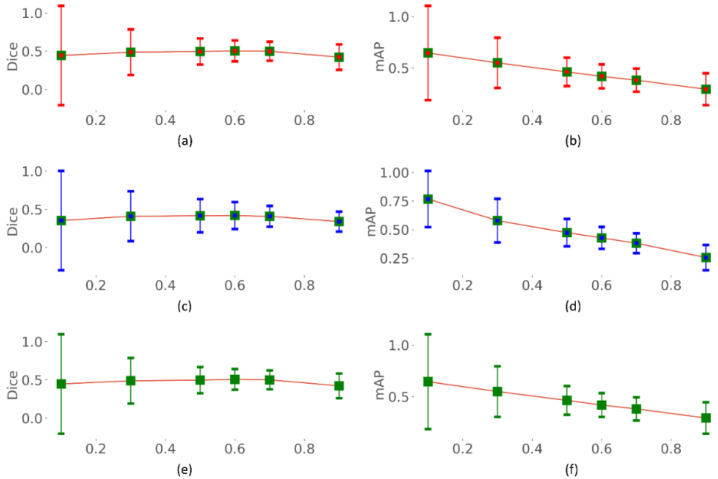
Dice index and mAP measured for varying values of *τ*. (**a**,**b**) using aleatoric uncertainty; (**c**,**d**) using epistemic uncertainty; and (**e**,**f**) using total (aleatoric + epistemic) uncertainty.

**Figure 9 biomedicines-10-01323-f009:**
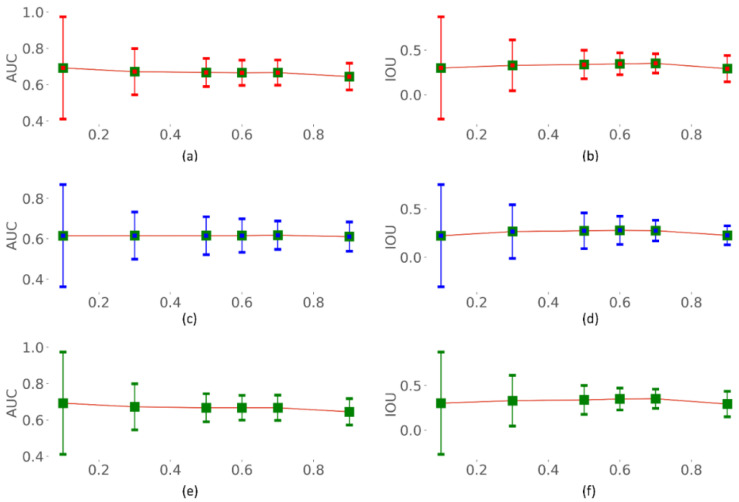
AUC and IOU measured for varying values of *τ*. (**a**,**b**) using aleatoric uncertainty; (**c**,**d**) using epistemic uncertainty; and (**e**,**f**) using total (aleatoric + epistemic) uncertainty.

**Figure 10 biomedicines-10-01323-f010:**
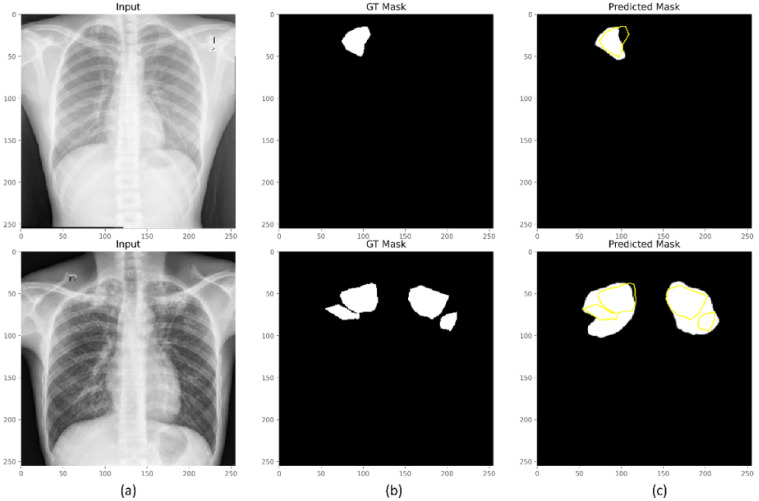
Predictions of the VGG-16-based U-Net model trained with the Focal Tversky + BU loss for sample CXR instances with unknown and other labels. (**a**) Input CXRs; (**b**) GT masks, and (**c**) predicted masks. The region highlighted in yellow denotes the contour of the GT masks.

**Table 1 biomedicines-10-01323-t001:** Dataset and its respective patient-level train/validation/test splits. The value *n* denotes the total number of CXRs in the collection that shows radiological signs of TB.

Dataset	Train	Validation	Test
Shenzhen TB (*n* = 330)	2231	66	33

**Table 2 biomedicines-10-01323-t002:** Test performance achieved by the VGG-16-based U-Net model trained with the proposed loss functions. Bold numerical values denote superior performance. Values in parenthesis denote the 95% CI for the mAP metric.

Loss	AUC	*mAP*	*IOU*	Dice
CE	0.7763	0.5150 (0.3444, 0.6856)	0.3334	0.5000
CE + BU	0.7704	0.5289 (0.3585, 0.6993)	0.3511	0.5197
Dice	0.7961	0.5497 (0.3799, 0.7195)	0.3653	0.5351
Dice + BU	0.7359	0.5162 (0.3456, 0.6868)	0.3400	0.5074
IOU	0.7864	0.5248 (0.3544, 0.6952)	0.3398	0.5073
IOU + BU	0.7477	0.5337 (0.3634, 0.7040)	0.3563	0.5254
Tversky	0.8075	0.5366 (0.3664, 0.7068)	0.3405	0.5080
Tversky + BU	0.8125	0.5368 (0.3666, 0.7070)	0.3364	0.5034
Focal Tversky	**0.8284**	0.5400 (0.3699, 0.7101)	0.3242	0.4896
Focal Tversky + BU	0.8281	**0.5710 (0.4021, 0.7399)**	**0.3723**	**0.5426**

**Table 3 biomedicines-10-01323-t003:** Performance comparison using the baseline predictions and prediction averaging with 30 MC forward passes. Bold numerical values denote superior performance.

Model	AUC	*mAP*	*IOU*	Dice
Focal Tversky + BU (Baseline)	**0.8281**	0.5710 (0.4021,0.7399)	0.3723	0.5426
Monte–Carlo (30) Focal Tversky + BU	0.8279	**0.5721 (0.4032, 0.7410)**	**0.3726**	**0.5430**

## Data Availability

The minimal data required to reproduce this study are available in terms of the figures, performance metrics, and other measures reported in the tables. The ground truth masks will be released in our forthcoming study.

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
