# Peer review of "Uncertainty Quantification in Segmenting Tuberculosis-Consistent Findings in Frontal Chest X-rays"

_biomedicines, 2022, doi:10.3390/biomedicines10061323_

Round 1

Reviewer 1 Report

Thank you for giving me an opportunity to review this outstanding paper. It is a well-conducted and written paper. However, there are several issues that need to be done. 

  1. Please separate the results and discussion part because it can make the reader confused.
  2. Please write principle findings in the discussion part and give a more clinical perspective.
  3. Please mention the strengths and limitations of this study.
  4. In conclusion, please mention how these findings can be used in real-world clinical settings.

Author Response

We render our sincere thanks to the reviewer for the insightful comments and for encouraging the resubmission of our manuscript. To the best of our knowledge and belief, we have addressed the concerns of the reviewer in the revised manuscript.

Q1: Please separate the results and discussion part because it can make the reader confused.

Author response: Agreed. The results (Section 3) and related discussions (Section 4) are separated to convey clarity to the readers.  

Q2: Please write principle findings in the discussion part and give a more clinical perspective.

Author response: Agreed. The changes are made to a part of the discussion section as shown below:

This study, however, suffers from the limitation that the CXRs manifesting TB-consistent lesions that are used to train and evaluate the segmentation models are limited. Addition-al diversity in the training process could be introduced by using CXR data and annotations from cross-institutions. Next, novel model optimization methods and loss functions can be proposed to further minimize prediction uncertainty and improve confidence. It is to be noted that more recent architectures like SegNet [32] and Trilateral attention net [33] could have been used in this study. However, the objective of this study is not to propose a new model or demonstrate state-of-the-art results for TB-consistent region segmentation. Rather, it is to validate the use of appropriate loss functions suiting the data under study and quantify uncertainty in model representations. Any DL model architecture could be modified to establish Bayesian variational inference using the MCD method for uncertainty quantification. These models that provide predictive estimates of uncertainty could be widely integrated into clinical practice to flag clinicians for alternate opinions, thereby imparting trust, and improving patient care. Research is ongoing in proposing novel methods for quantifying and explaining uncertainties in model predictions [34,35]. With the advent of high-performance computing and storage solutions, several models with deep and diverse architectures can be trained to construct ensembles with reduced prediction uncertainty and deployed in the cloud to be used for real-time clinical applications. This study, however, serves as a paradigm to quantify uncertainties and establish an uncertainty threshold for segmenting disease-specific ROIs in CXRs and could be extended to other medical imaging modalities. Future studies will focus on quantifying uncertainties in multiclass and multi-label medical data classification, regression, segmentation, and uncertainty-based active learning.

Q3: Please mention the strengths and limitations of this study.

Author response: We wish to reiterate our response to Q2.

Q4: In conclusion, please mention how these findings can be used in real-world clinical settings.

Author response: Agreed. In the revised manuscript, we have discussed how the current proposal could be used in real-world settings. These are shown in the authors' responses for Q2.

Reviewer 2 Report

The paper addresses an interesting area or study. However, there are certain issues which currently raise some concerns regarding the presented study:

1) The notation ‘p’ is used in too many contexts
. For a reader not strictly specialized in the field, this paper is very difficult to be understood. Some of them can be confusing, such as

P for p-value, p for pixel, p^ for activation function, p and p’ for probability distributions in the case of GT and prediction, P for output from the Softmax layer, p for Sigmoid function. I suggest the authors to change the notations for a clear description.

2) The same for P (Precision) or Precision/ Recall or R. See eq. 19, 20 and 21.

3) Fig. 5. To be able to estimate the computational cost the authors should clarified the unit of the computation time.

4) Fig. 6. It is unclear where/which the c) and d) samples are

5) Lines 382-383. The statement “Regions with lower uncertainties look “unfilled” while those 382 with greater uncertainties appear prominent” would be better understood if arrows are inserted in the images

6) For a clear representation of the augmentation technique, some images should be displayed.

7) Are there any limitations to the proposed method?

8) Some references are outdated. They should be updated.

Author Response

The paper addresses an interesting area or study. However, there are certain issues which currently raise some concerns regarding the presented study:

Author response: We render our sincere thanks to the reviewer for these valuable comments and for encouraging the resubmission of our manuscript.

Q1: The notation ‘p’ is used in too many contexts. For a reader not strictly specialized in the field, this paper is very difficult to be understood. Some of them can be confusing, such as P for p-value, p for pixel, p^ for activation function, p and p’ for probability distributions in the case of GT and prediction, P for output from the Softmax layer, p for Sigmoid function. I suggest the authors to change the notations for a clear description. The same for P (Precision) or Precision/ Recall or R. See eq. 19, 20 and 21.

Author response: We regret the lack of clarity in the use of notations. We have changed the notations for all the equations suggested by the reviewer to improve readability.

Q2: Fig. 5. To be able to estimate the computational cost the authors should clarified the unit of the computation time.

Author response: Agreed. We computed pixel-wise uncertainties where we investigated the mean and variance of the uncertainty metrics including aleatoric, epistemic, and entropy. The units of entropy depend on the base (b) of the logarithm used. Here, we used b=2, in this case, the units of entropy are expressed in Shannon/bits. The elapsed time is measured in seconds. The length of an error bar indicates the uncertainty of the measured metric. The error cap highlights where exactly an error bar ends. These are mentioned in the revised manuscript. During inference, we perform N forward passes through the trained model. The value of N ranges from 1 to 50. During each forward pass, a different set of model units are sampled to be dropped out which results in stochastic predictions that can be interpreted as samples from a probabilistic distribution. The uncertainty in the model predictions is quantified using the probabilistic samples during N forward passes. The aforementioned uncertainty metrics are plotted against the number (N) of Monte-Carlo forward passes.

Q3: Fig. 6. It is unclear where/which the c) and d) samples are

Author response: Agreed. We regret the typos in the initial submission and modified Figure 6 in the revised manuscript.

Q4: Lines 382-383. The statement “Regions with lower uncertainties look “unfilled” while those 382 with greater uncertainties appear prominent” would be better understood if arrows are inserted in the images.

Author response: Agreed. Figure 6 in the revised manuscript is modified by adding a blue bounding box showing sample predicted pixels with high uncertainty, and a red bounding box showing those with low uncertainty.

Q5: For a clear representation of the augmentation technique, some images should be displayed.

Author response: Agreed. A new figure (Figure 1) showing augmented samples for a CXR instance is added to the revised manuscript.

Q6: Are there any limitations to the proposed method?

Author response: Yes. The limitations of the study and future directions are included in the “Discussion and Conclusion” section of the revised manuscript as given below:  

This study, however, suffers from the limitation that the CXRs manifesting TB-consistent lesions that are used to train and evaluate the segmentation models are limited. Addition-al diversity in the training process could be introduced by using CXR data and annotations from cross-institutions. Next, novel model optimization methods and loss functions can be proposed to further minimize prediction uncertainty and improve confidence. It is to be noted that more recent architectures like SegNet [32] and Trilateral attention net [33] could have been used in this study. However, the objective of this study is not to propose a new model or demonstrate state-of-the-art results for TB-consistent region segmentation. Rather, it is to validate the use of appropriate loss functions suiting the data under study and quantify uncertainty in model representations. Any DL model architecture could be modified to establish Bayesian variational inference using the MCD method for uncertainty quantification. These models that provide predictive estimates of uncertainty could be widely integrated into clinical practice to flag clinicians for alternate opinions, thereby imparting trust, and improving patient care. Research is ongoing in proposing novel methods for quantifying and explaining uncertainties in model predictions [34,35]. With the advent of high-performance computing and storage solutions, several models with deep and diverse architectures can be trained to construct ensembles with reduced prediction uncertainty and deployed in the cloud to be used for real-time clinical applications. This study, however, serves as a paradigm to quantify uncertainties and establish an uncertainty threshold for segmenting disease-specific ROIs in CXRs and could be extended to other medical imaging modalities. Future studies will focus on quantifying uncertainties in multiclass and multi-label medical data classification, regression, segmentation, and uncertainty-based active learning.

Q7: Some references are outdated. They should be updated.

Author response: Agreed. We thank the reviewer for these valuable comments. We have included recent studies on uncertainty measurements in the revised manuscript as shown below:  

  1. Sagar, A. Uncertainty Quantification Using Variational Inference for Biomedical Image Segmentation. Proc. - 2022 IEEE/CVF Winter Conf. Appl. Comput. Vis. Work. WACVW 2022 2022, 44–51, doi:10.1109/WACVW54805.2022.00010.
  2. Tang, P.; Yang, P.; Nie, D.; Wu, X.; Zhou, J.; Wang, Y. Unified Medical Image Segmentation by Learning from Uncertainty in an End-to-End Manner. Knowledge-Based Syst. 2022, 241, doi:10.1016/j.knosys.2022.108215.

Round 2

Reviewer 1 Report

Thanks for the revised version.  It can be considered for publication.